# Prevalence of sending, receiving and forwarding sexts among youths: A three-level meta-analysis

Cristian Molla-Esparza[1]*, Josep-Maria Losilla[2], Emelina López-González[1]

1 Department of Research Methods and Educational Diagnosis, University of Valencia, UVEG, Valencia, Spain, 2 Department of Psychobiology and Methodology of Health Science, Autonomous University of Barcelona, UAB, Barcelona, Spain

☯ These authors contributed equally to this work.

* Cristian.Molla@uv.es

**Data Availability Statement:** All relevant data are within the manuscript and its Supporting Information files.

## Abstract

By systematic review with a three-level, mixed-effects meta-analysis, this paper examines the prevalence of sexting experiences among youths aimed at analyzing conceptual and methodological moderators that might explain its heterogeneity. A search was conducted of five bibliographic databases and grey literature up until February 2020. The risk of bias in primary studies was assessed. A total of seventy-nine articles met the set inclusion criteria. Mean prevalences for sending, receiving and forwarding sexts were .14 (95% CI: .12, .17), .31 (95% CI: .26, .36) and .07 (95% CI: .05, .09), respectively, expressed as fractions over one. Moderator analyses showed that all sexting experiences increased with age (e.g., the mean prevalence for sending sexts at the age of 12 was .04, whereas, at the age of 16, it was .21) and year of data collection (e.g., the mean prevalence for sending sexts in studies collecting data in 2009 was .07, whereas, in studies collecting data in 2018, it was .33). Sub-group analysis revealed that studies with probabilistic samples led to significantly lower mean prevalences for the sexting experiences of sending (.08, 95% CI: .06, .11), receiving (.19, 95% CI: .15, .24) and forwarding sexts (.04, 95% CI: .03, .07). Self-reported administration procedures also led to more homogeneous prevalence estimates than interviews. Prevalence estimates also varied according to the type of media content (e.g., the mean prevalence for sending sexual text messages was .22, whereas, for sending sexual images or videos, it was .12). Overall, our meta-analysis results suggest high and increasing prevalences of sending and receiving sexts among youths.

## Introduction

Sexting, generally defined as the sending, receiving or forwarding of erotic or sexual media content (messages, photos or videos), through interactive technological devices, mainly mobile devices, is prevalent among youths [1, 2]. In the last few years, sexting has gained increasing empirical attention due to its implications and possible consequences. A part of the research literature frames sexting as a normalized and legitimate sexual activity that allows youths to

**Funding:** This work was supported by funding from the Government of the Valencian Community (predoctoral grant DOGV No. 7943, ACIF, 837 2017) and from the Spanish Ministry of Science, Innovation and Universities (PGC2018-100675-B-I00). CME also thanks the Government of the Valencian Community (DOGV No. 7943) for financial support for his research stay with the Meta-analysis Team of the University of Murcia (Spain), coordinated by Dr. JSM. Funders had no role in the study design, data collection or analysis, the decision to publish, or the preparation of the manuscript.

**Competing interests:** The authors have declared that no competing interests exist.

satisfy certain needs relating to the exploration and discovery of their own sexual identity, and the initiation or maintenance of new affective or sexual relationships [3, 4]. However, the available empirical evidence also suggests that sexting entails risks such as the intentional, non-consensual distribution of sexts beyond the intended recipient [5]. Several empirical studies have also found that sexting involvement was associated with participation in undesirable dynamics such as dating violence, sextortion, cyberbullying and grooming [6–10]. Likewise, in some studies, sexting has also been associated with anxiety and depression symptomology, as well as attempted or ideated suicide [11–13].

Sexting prevalence rates observed in youths indicate great variability, and demographic correlates are inconclusive, especially concerning gender differences [1, 2, 14]. To date, a number of studies have examined sexting prevalence rates among youths. Klettke et al. [1] analyzed 12 studies with samples of adolescents under 19 years old, obtaining a mean prevalence of sending and receiving sexts of 10% (95% CI: 2%, 19%) and 16% (95% CI: 12%, 20%), respectively, with a large confidence interval of means. This review [1] also conclude that studies with non-probabilistic samples obtained higher point prevalence estimates compared to those with probabilistic samples. Also, the prevalence of sending and receiving erotic content appeared to be lower among youths than among adults. More recently, Madigan et al. [2] contributed to the field by conducting a meta-analysis of 39 studies with participants under 18 years old, obtaining mean prevalences for sending (from 34 studies), receiving (from 20 studies) and forwarding (from 5 studies) sexts of 15% (95% CI: 13%, 17%), 27% (95% CI: 23%, 32%) and 12% (95% CI: 8%, 16%), respectively, again with a high variability in results ($I^2$ = 98% to 99%, respectively). Madigan et al. [2] also showed that prevalence rates were higher among older youths and that they increased over time. Furthermore, they found that rates of sexting were not moderated by publication status (e.g. peer reviewed vs. dissertation/report) or geographical location. In addition, both the aforementioned reviews [1, 2] agreed that the prevalence of receiving sexts was higher than the prevalence of sending sexts. Both reviews also agreed in proposing further study of conceptual aspects such as distinguishing between different media formats and degrees of explicitness of the exchanged contents. Thus far, the Madigan et al. [2] study has been the only review that has elaborated a meta-estimate of the sexting prevalence among youths.

The review study carried out by Barrense-Dias et al. [15] noted that definitions of sexting among studies differ in elements such as the actions the practice of sexting entails, the different types of media content transmitted, the degree of sexual explicitness of the content, the timeframe of the measure, and the context in which sexting is practiced. For example, while some studies have focused on asking about the sending of nude pictures to romantic partners without indicating a temporal timeframe [16], others have asked whether during the last twelve months prior to the survey participants have received sexual text messages, images or videos without defining the context in which the action was carried out [17, 18].

Given the great heterogeneity encountered in prevalence estimates and the growing trend of this risky behavior over time, we considered it opportune to conduct a new meta-analysis. Therefore, the first aim of this research was to update the previous meta-analytic synthesis on sexting prevalence among youths [2]. The second aim was to identify and analyze new potential moderators in terms of methodological aspects (e.g., the sampling techniques and administration procedure used) and conceptual aspects (e.g., the degree of sexual explicitness of the media content, the context in which sexting is practiced, the willingness of participants or the timeframe of the measure) that may explain the observed heterogeneity in sexting prevalences. The present study also adds to the current literature by applying a state-of-the-art, three-level meta-analytic approach to estimating the mean prevalence of sexting experiences, considering the dependence among multiple sexting experiences from the same study. The ultimate goal is to contribute to the development of consensus on a clear definition of sexting.

## Method

A systematic review and meta-analysis were carried out following the methodology of 'Preferred Reporting Items for Systematic Reviews and Meta-Analyses' (PRISMA) [19, 20].

## Document search and selection

A search was carried out between October 2019 and February 2020, resulting in the selection of the following databases: Education Resources Information Center (ERIC), via ProQuest; Psychological Information (PsycINFO), via APA PsycNET; Medical Literature Analysis and Retrieval System (MEDLINE), via ProQuest; Scopus, via Elsevier; and ISI Web of Science (WoS CORE Collection), via Thomson Reuters. The search strategy followed the 'Peer Review of Electronic Search Strategies' (PRESS) guideline [21]. The term 'sexting' used in previous review studies [1, 2, 4, 11, 15, 22–24] was applied as a descriptor in order to identify a significant number of studies originating in various scientific fields, such as those of Psychology, Education, Sociology, Technology, Health Sciences and Legal Sciences. In order to provide a more comprehensive review, a 'gray literature' search was carried out using the Google and Google Scholar search engines with the following terms: "sexting", "sext", "sexual texting" and "sexual messaging". Weekly alerts were programmed for new research in PsycINFO, Web of Science, and Scopus, up until March 10th, 2020. The reference lists of relevant empirical articles and reviews were also checked to identify other potentially eligible studies. Additionally, we contacted corresponding authors via e-mail and/or ResearchGate to request full-texts or to gather additional information on their studies (6 out of 17 solicitations were answered, and 3 met our requests). To facilitate replication of this review, S1 Table contains the specific search strategy used in each database consulted.

## Inclusion and exclusion criteria

In accordance with the stated objectives of this research, studies were included if they: a) aimed to examine the prevalence of sexting and/or its correlates; b) comprised a sample of participants up to 18 years old; c) provided original empirical data; and d) were available in English or Spanish. Regarding the inclusion criterion a), three possible prevalence percentages were considered in relation to each study, corresponding respectively to the specific actions of: sending; receiving; and forwarding.

First, articles meeting the inclusion criteria were selected, and, when decisions could not be made from the title and abstract alone, the full paper was retrieved as well. The selected papers were checked independently by the authors CME and ELG. Any discrepancies were resolved through discussion with a third author (JML) where necessary.

Different studies analyzing data from the same research project were included only when the sample or the measure of sexting differed among them. S2 Table summarizes the excluded studies, while Fig 1 illustrates the flowchart of the systematic review process. The studies included in the meta-analysis are referenced in S1 Appendix.

## Data coding

For coding purposes, the following data, including bibliometric information and the research strategies of the original studies, was recorded: a) type of publication (degree or master thesis, article or report, peer-reviewed, not peer-reviewed or under review); b) year of publication of the study; c) year of data collection; d) geographical origin of the samples classified according to seven-continent model: Africa, Asia, Europe, North America, South America, Antarctica and Oceania/Australia; e) study design (cross-sectional or longitudinal survey); f) type of

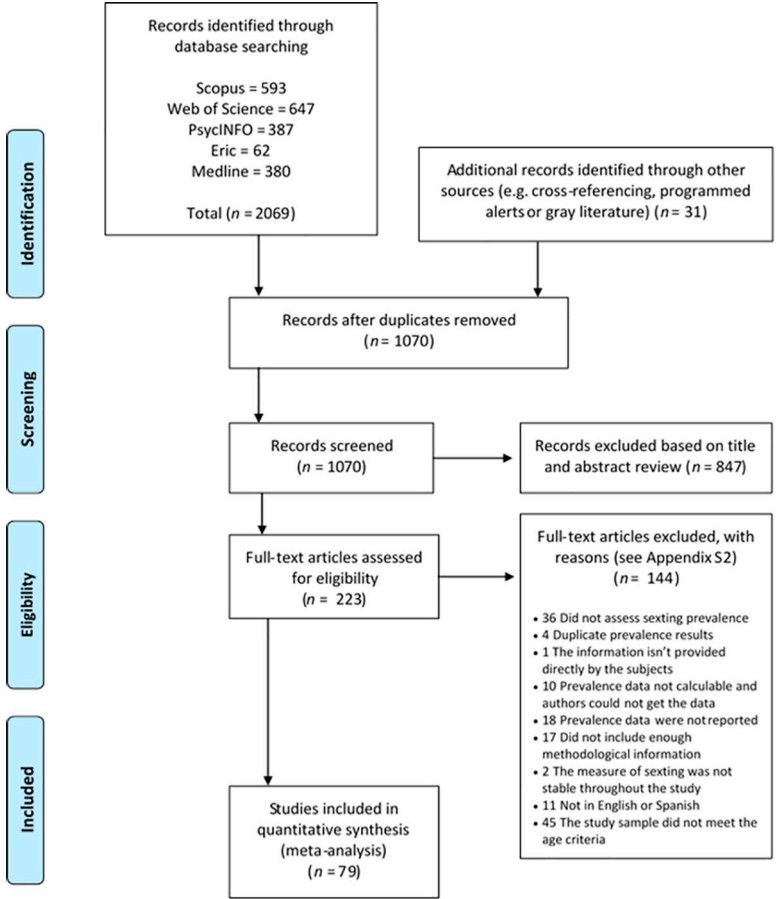

**Fig 1. PRISMA flowchart of the study selection process.**

sample (probabilistic or non-probabilistic); g) reference population (e.g., preadolescents, middle school students, high school students); h) sample size and proportion of women; i) range, mean and standard deviation of the age of the participants; j) administration procedure (telephone or face-to-face interview, online, paper-based or mixed survey); k) message content (text messages, images/videos, or both); l) degree of sexual explicitness of the content (nude, not nude, both); m) context in which sexting is practiced (romantic relationship, others or not defined); n) willingness of the participants in sexting actions (sending: voluntary, not voluntary, not defined; receiving: solicited, unsolicited, not defined; forwarding: with consent, without consent, not defined); o) timeframe of the measure of sexting ($\leq$ 6 months or $>$ 6 months, lifetime, or not defined); and p) sexting action prevalence results (sending, receiving, forwarding).

In certain cases, additional calculations were made to determine percentages. In addition, when a study was longitudinal, only the prevalence rate of the first timeframe was recorded.

## Study quality assessment

A critical appraisal of the studies (see S3 Table) was performed using a tool elaborated by the authors based on that proposed by the National Institute of Health and Care Excellence (NICE) [25] for prognostic studies. This tool evaluates five methodological quality domains: a) study design (cross-sectional or longitudinal survey); b) sampling technique (probabilistic or

non-probabilistic); c) sexting measure quality (evidence of validity and reliability in the study sample or in comparable samples, same or equivalent measure procedure for all participants, and non-significant proportion of non-responses); d) timeframe of the sexting behaviors (well defined or undefined); and e) response rate (calculated by dividing the number of participants completing the survey by the number of solicited participants).

Data extraction and quality assessment of the included studies were performed by the authors CME, JML and ELG. Any discrepancies regarding data extraction and quality assessment of the included studies were resolved through consensus. The potential effect of study quality on prevalence values was assessed and indicated in the results tables.

## Analysis

The meta-analysis was conducted using multilevel, linear, random and mixed effects models in order to estimate the mean prevalences of sexting experiences, with associated 95% confidence intervals (CI) and credibility intervals (CRs) around the estimates. In particular, the adjusted three-level, meta-analytic model featured variance components distributed as follows: a sampling variation for each effect size at level one; a variation over outcomes within a study at level two; and a variation over studies at level three [26, 27]. Unlike the traditional two-level, univariate approach, this three-level strategy is more efficient since it allows all data from studies with multiple outcomes to be analyzed simultaneously, taking into account the dependence among effect sizes from the same study, opportune in the case of studies about sexting prevalence which usually report the prevalence of various sexting experiences (i.e., sending, receiving, and forwarding sexts). By ignoring the dependence in effect sizes, the two-level model can result in standard errors that are too small, and therefore in largely deflated coverage proportions of confidence intervals [28]. Furthermore, the application of a three-level meta-analysis is especially appropriate when the outcomes of interest vary in measurement form across studies [29].

All prevalence rates were transformed into logit event rate effect sizes before the analysis, and the results were retransformed into fractions over 1 in order to facilitate ease of interpretation. $Q$ and $Tau^2$ statistics were computed to assess the statistical heterogeneity of effect sizes. Between-study heterogeneity was also examined using $Q$ statistic (categorical moderators) and meta-regressions (quantitative moderators) [30]. Specific functions were used to examine a) profile likelihood plots of the variance components, b) potential outlying and influential studies and/or outcomes, and c) potential publication bias. No data points had a Cook's distance exceeding the cut-off value of 3 standard deviations (SD). Studies with the highest studentized residuals and Cook's D values (Maheux et al. [31] and Fix et al. [32] for sending sexts, and Gewirtz-Meydan et al. [33] and Mitchell et al. [34] for receiving sexts) were retained from the original model because of their limited influence (with small weights ranging from .26% to .27%,) and also because, after reviewing these studies in detail, we found no reasons to exclude them.

All analyses were carried out with the Metafor package (version 2.4–0) for R [35]. Relevant R code and graphs are provided in S2 Appendix.

## Results

### Search results

The initial systematic literature search yielded 2069 potentially eligible studies. A further 31 studies were subsequently added from cross-referencing, programmed alerts and the gray literature search. After duplicates had been eliminated, 1070 studies remained, of which 991 were excluded on the basis of their titles, abstracts or content (see Fig 1 and S2 Table).

Consequently, a total of 79 articles relating to sexting prevalence were included in the final meta-analysis and quality assessment.

The documents analyzed were predominantly articles published in scientific journals and subject to the peer-review process ($n = 71$, 90%). Most of the studies reporting sexting prevalence were conducted in the United States ($n = 34$, 43%) and Spain ($n = 11$, 14%) (S4B Table contains detailed information on the geographical origin of the samples). More than half ($n = 48$, 61%) were published between 2016 and 2020. The most commonly used tools to measure prevalence were questionnaires, employed online ($n = 20$, 27%) or on paper ($n = 36$, 48%), followed by telephone or face-to-face interviews ($n = 6$, 8%), and mixed online and paper surveys ($n = 5$, 7%) (Table 1). The included studies involved a total of 184695 participants. Finally, in all studies reporting prevalences of sending and receiving sexts, subjects received more sexts than they sent (detailed information on the studies included is provided in S4A Table).

## Study quality and methodological moderators

The quality assessment revealed that almost all the studies analyzed were cross-sectional studies ($n = 71$, 90%) (Table 1). Most used non-probabilistic sampling techniques ($n = 51$, 65%). As indicated in Table 3, analysis of the sampling techniques applied in the studies revealed statistically significant differences in prevalence estimates of the sexting experiences of sending, receiving and forwarding ($Q_M (3) = 32.88$, $p < .01$). Lower prevalences were obtained from probabilistic samples (.08, 95% CI: .06, .11; .19, 95% CI: .15, .24; and .04, 95% CI: .03, .07) than from non-probabilistic ones (.19; 95% CI: .15, .22; .39, 95% CI: .34, .45; and .10, 95% CI: .07, .13, respectively). Regarding the quality of the measure, only 19% ($n = 15$) of the included studies reported any reliability index or evidence of the validity of the sexting measures applied. In the case of forwarding sexts, studies classified with a low risk of bias indicated a significantly higher estimate (.15; 95% CI: .09, .25) than studies classified with a significant risk (.06; 95% CI: 04, 07) ($Q_M (3) = 18.67$, $p < .01$).

Additionally, 76% of the studies ($n = 60$) provided no information on the response rate. Among the studies that reported such information ($n = 19$, 24%), participation was generally low: more than half ($n = 11$) reported $\leq 60\%$ of solicited participants responding, whereas only four studies reported $\geq 80\%$ responding. Lastly, regarding the experience of forwarding

**Table 1. Summary of the critical appraisal of studies included in the review.**

| | | Studies ($n = 79$) |
|---|---|---|
| | | $n$ (%) |
| Study design | Cross-sectional | 71 (90%) |
| | Longitudinal | 8 (10%) |
| Sampling technique | Probabilistic | 28 (35%) |
| | No probabilistic | 51 (65%) |
| Q. Measurement (Risk of bias) | Low risk | 15 (19%) |
| | Significant risk | 57 (72%) |
| | Insufficient information | 7 (9%) |
| Temporal framework | Well defined | 34 (43%) |
| | Lifetime or undefined | 45 (57%) |
| Response rate | Reported | 19 (24%); IQR: 25.70%– 76%; M: 45.04%. |
| | Not reported | 60 (76%) |

"IQR" = Interquartile range, "M" = Median.

sexts, analysis of the timeframe of the measure revealed statistically significant differences in its prevalence estimates ($Q_M$ (3) = 9.54, $p$ = .02 and $p$ = .01 for this experience). Studies evaluating the prevalence of forwarding sexts in timeframes equal to or less than 6 months reported significantly lower prevalences (.03; 95% CI: .02, .06) compared to studies without timeframes or with indicated timeframes exceeding six months (.08; 95% CI: .06, .11). In summary, most of the studies considered were cross-sectional and non-probabilistic, with low or unreported response rates and poor measure quality.

## Sexting prevalence and conceptual moderators

As indicated in Table 2, the analysis of the differences between sexting experiences revealed relevant and statistically significant differences ($Q_M$ (3) = 681.28, $p$ < .01). Receiving sexts had a considerably higher global prevalence (.31; 95% CI: .26, .36) than sending sexts (.14; 95% CI: .12, .17) and forwarding sexts (.07; 95% CI: .05, .09). These prevalences increased over time ($Q_M$ (3) = 23.13, $p$ < .01), with the trend showing, for example, that sending sexts in studies collecting data in 2009 gave .07 (95% CI: .05, .10), whereas studies collecting data in 2018 gave .33 (95% CI: .22, .46). The same trend was also observed in receiving and forwarding experiences (Table 2).

**Table 2. Overall mean prevalences of sending, receiving and forwarding sexts by year of data collection.**

|  | $K$ | $eff$ | (95% CI) | (95% CRs) | $Tau^2$ | Overall prevalences: Test of Residual Heterogeneity and Moderators | Year of data collection: Test of Residual Heterogeneity and Moderators |
|---|---|---|---|---|---|---|---|
| Sending | 57 | .14 | (.12, .17) | (.03, .47) | .73 | $Q_E$ (106) = 12232.15, $p$ < .01 | $Q_E$ (64) = 7053.10, $p$ < .01 |
| 2009 | 5 | .07 | (.05, .10) | (.02, .27) |  | $Q_M$ (3) = 681.28, $p$ < .01 | $Q_M$ (3) = 23.13, $p$ < .01 |
| 2014 | 16 | .16 | (.13, .20) | (.04, .48) |  |  |  |
| 2018 | 14 | .33 | (.22, .46) | (.09, .71) |  |  | For sending sexts $p$ < .01 |
| Receiving | 39 | .31 | (.26, .36) | (.08, .70) | .69 |  |  |
| 2009 | 4 | .16 | (.11, .23) | (.04, .50) |  |  |  |
| 2014 | 15 | .34 | (.28, .41) | (.10, .72) |  |  |  |
| 2018 | 10 | .58 | (.43, .71) | (.20, .88) |  |  | For receiving sexts $p$ < .01 |
| Forwarding | 13 | .07 | (.05, .09) | (.01, .30) | .76 |  |  |
| 2009 | 1 | .03 | (.01, .07) | (.00, .15) |  |  |  |
| 2014 | 1 | .08 | (.05, .12) | (.02, .30) |  |  |  |
| 2018 | 4 | .20 | (.09, .37) | (.04, .60) |  |  | For forwarding sexts $p$ < .01 |

"$k$" = number of studies included, "$eff$" = effect size (prevalence), "95% CI" = 95% confidence interval, "95% CRs" = 95% credibility intervals, "$Q_E$" = within-categories statistic to test the model misspecification, "$Q_M$" = between-categories statistic to test the influence of the moderator variable on the prevalence rates, "$Tau^2$" = Residual heterogeneity for the levels of the inner factor, "$p$" = p-values for the test statistics.

Note 1: To estimate overall prevalence and make subsequent calculations regarding studies that reported more than one rate, we used the closest at the time of data collection.

Note 2: Among the included studies reporting the year of data collection, none reported a year beyond 2018.

**Sending sexts.** As indicated in Table 3, moderator analysis showed that the mean age of study participants was positively related to the prevalence of sending sexts ($Q_M$ (3) = 148.00, $p$ < .01). The prevalence of sending sexts at the age of 12 was .04 (95% CI: .02, .06), at the age of 14 was .09 (95% CI: .07, .12), and at the age of 16 was .21 (95% CI: .17, .25). The same trend was also observed in receiving and forwarding experiences (Table 3). The observed prevalences and the overall mean estimate of sending sexts are depicted in Fig 2A.

Significant differences were also identified in the types of media content transmitted ($Q_M$ (2) = 366.07, $p$ < .01). The sending of text messages obtained a significantly higher global prevalence (.22; 95% CI: .18, .27) than the sending of pictures or videos (.12; 95% CI: .10, .15). Prevalence was not moderated by the type of publication, the administration procedure, the risk of bias in the measure of sexting, the timeframe of the measure of sexting, the context in which sexting was practiced, the degree of sexual explicitness of the content, the willingness of participants, their sex or the geographical origin of samples.

**Receiving sexts.** As in sending sexts, moderator analysis revealed that the prevalence of receiving sexts increased with the sample's mean age ($p$ < .01). The employed administration procedure showed a statistical relationship with prevalence rates ($Q_M$ (2) = 7.28, $p$ = .03 and $p$ < .01 for receiving sexts), with the highest prevalence rates when the studies used self-reported administration procedures (.34; 95% CI: .29, .39).

The remaining moderator variables in Table 3 did not indicate a significant relation. The observed prevalences and the overall mean estimate of receiving sexts are depicted in Fig 2B.

**Forwarding sexts.** Moderator analyses revealed that the prevalence of forwarding sexts increased with age ($p$ = .05). The remaining moderator variables in Table 3 did not have a significant relation. The observed prevalences and the overall mean estimate of forwarding sexts are depicted in Fig 2C.

## Discussion

This systematic review and meta-analysis research examines the prevalence of sexting experiences via three-level, mixed-effects, meta-analysis models. In addition, it provides an updated meta-estimate of the prevalence of sexting experiences among youths, analyzing a wide range of methodological and conceptual factors susceptible to moderating the heterogeneity of results reported in the empirical literature. Regarding conceptual factors, a differentiating contribution of this research is the classification and analysis of the moderating effects on sexting prevalence of new key elements in sexting's operational definition: the degree of sexual explicitness of the content, the background context to the sexting, the willingness of participants, and the timeframe of the sexting measure.

The results obtained in this research reveal that the prevalence estimate of sending sexts is consistent with those reported in previous reviews [1, 2], with overlapping confidence intervals providing good evidence of concurrent validity. The prevalence estimate of receiving sexts in this research is consistent with that reported by Madigan et al. [2], but is significantly higher than reported by Klettke et al. [1]. Lastly, the estimated prevalence of forwarding sexts also coincides with that reported by Madigan et al. [2], although our estimate is slightly lower. However, considering that the practice of sexting is more prevalent over time, the more relevant prevalences reported in our study may be those stratified by year of data collection, especially those clustered in recent years. Indeed, in these years, the prevalence estimates of sending, receiving and forwarding sexts were significantly higher than the average prevalence estimates pooling all the studies reviewed, and also greater than the overall mean prevalence estimates reported by Klettke et al. [1] and Madigan et al. [2]. Finally, in accordance with previous reviews, our meta-analysis revealed a high dispersion

**Table 3. Results of the three-level, meta-regression analyses with moderators of the prevalences of sending, receiving and forwarding sexts.**

| | Sending | | | | | Receiving | | | | | Forwarding | | | | | Comparison |
|---|---|---|---|---|---|---|---|---|---|---|---|---|---|---|---|---|
| | k | eff | (95% CI) | Tau² | p | k | eff | (95% CI) | Tau² | p | k | eff | (95% CI) | Tau² | p | |
| Document type | | | | .73 | .70 | | | | .69 | .95 | | | | .95 | .25 | $Q_E$ (103) = 12085.73, $p < .01$ |
| Not peer-reviewed | 8 | .13 | (.07, .21) | | | 8 | .31 | (.20, .44) | | | 3 | .09 | (.05, .18) | | | $Q_M$ (3) = 3.38, $p = .34$ |
| Peer-reviewed | 49 | .14 | (.12, .17) | | | 31 | .30 | (.26, .36) | | | 10 | .06 | (.04, .08) | | | |
| Sampling technique | | | | .53 | < .01 | | | | .43 | < .01 | | | | .55 | < .01 | $Q_E$ (103) = 7869.97, $p < .01$ |
| Non-probabilistic | 39 | .19 | (.15, .22) | | | 23 | .39 | (.34, .45) | | | 9 | .10 | (.07, .13) | | | $Q_M$ (3) = 32.88, $p < .01$ |
| Probabilistic | 18 | .08 | (.06, .11) | | | 16 | .19 | (.15, .24) | | | 4 | .04 | (.03, .07) | | | |
| Administration procedure | | | | .75 | .08 | | | | .56 | < .01 | | | | | | $Q_E$ (85) = 10532.89, $p < .01$ |
| Interview (PI or TI) | 3 | .07 | (.03, .15) | | | 6 | .18 | (.11, .28) | | | | | | | | $Q_M$ (2) = 7.28, $p = .03$ |
| Self-reported | 50 | .15 | (.12, .18) | | | 30 | .34 | (.29, .39) | | | | | | | | |
| Quality of the measure | | | | .74 | .08 | | | | .61 | .16 | | | | .57 | < .01 | $Q_E$ (92) = 10165.23, $p < .01$ |
| Significant risk | 41 | .12 | (.10, .15) | | | 28 | .28 | (.24, .33) | | | 10 | .06 | (.04, .07) | | | $Q_M$ (3) = 18.67, $p < .01$ |
| Low risk | 10 | .19 | (.12, .28) | | | 7 | .37 | (.26, .49) | | | 2 | .15 | (.09, .25) | | | |
| Temporality of the measure | | | | .74 | .71 | | | | .67 | .42 | | | | .51 | .01 | $Q_E$ (103) = 11709.47, $p < .01$ |
| < Six months | 9 | .13 | (.08, .20) | | | 5 | .27 | (.17, .38) | | | 3 | .03 | (.02, .06) | | | $Q_M$ (3) = 9.54, $p = .02$ |
| > Six months | 48 | .14 | (.12, .17) | | | 34 | .32 | (.27, .37) | | | 10 | .08 | (.06, .11) | | | |
| Geographical origin of samples [a] | | | | | | | | | | | | | | | | |
| Europe | 23 | .13 | (.10, .17) | | | 14 | .31 | (.24, .39) | | | 4 | .10 | (.05, .19) | | | $Q_E$ (38) = 7482.50, $p < .01$ |
| | | | | | | | | | | | | | | | | $Q_M$ (3) = 158.32, $p < .01$ |
| Spain | 6 | .16 | (.10, .25) | | | 7 | .29 | (.20, .41) | | | 3 | .14 | (.08, .21) | | | |
| Belgium | 4 | .16 | (.08, .30) | | | 2 | .27 | (.25, .28) | | | | | | | | |
| Netherlands | 2 | .11 | (.03, .34) | | | | | | | | | | | | | |
| Czech Republic | 4 | .16 | (.08, .30) | | | | | | | | | | | | | |
| Geographical origin of samples [a] | | | | | | | | | | | | | | | | |
| North America | 28 | .14 | (.10, .18) | | | 21 | .26 | (.20, .33) | | | 7 | .07 | (.04, .10) | | | $Q_E$ (53) = 3616.95, $p < .01$ |
| | | | | | | | | | | | | | | | | $Q_M$ (3) = 1327.73, $p < .01$ |
| Canada | 2 | .14 | (.13, .16) | | | 2 | .27 | (.26, .29) | | | | | | | | |
| Northern America | 25 | .17 | (.12, .22) | | | 18 | .25 | (.18, .33) | | | 6 | .08 | (.05, .11) | | | |
| South America | 2 | .26 | (.12, .47) | | | 3 | .43 | (.29, .58) | | | 1 | .18 | (.12, .25) | | | $Q_E$ (3) = 95.03, $p < .01$ |
| | | | | | | | | | | | | | | | | $Q_M$ (3) = 97.77, $p < .01$ |
| Ecuador | 2 | .26 | (.12, .47) | | | 2 | .48 | (.29, .67) | | | | | | | | |
| Asia | 2 | .22 | (.11, .37) | | | | | | | | | | | | | |
| Content of messages | | | | .62 | < .01 | | | | .54 | < .01 | | | | | | |
| Text | 6 | .22 | (.18, .27) | | | 2 | .37 | (.32, .43) | | | | | | | | $Q_E$ (78) = 5854.67, $p < .01$ |
| | | | | | | | | | | | | | | | | $Q_M$ (2) = 366.07, $p < .01$ |
| Images or videos | 37 | .12 | (.10, .15) | | | 27 | .27 | (.23, .32) | | | | | | | | |
| Explicitness of images / videos [b] | | | | | | | | | | | | | | | | |
| Nude | 10 | .15 | (.11, .21) | | | 10 | .30 | (.21, 42) | | | 5 | .09 | (.08, .10) | | | $Q_E$ (24) = 1438.82, $p < .01$ |
| | | | | | | | | | | | | | | | | $Q_M$ (3) = 2175.17, $p < .01$ |
| Not nude | 1 | .15 | (.11, .19) | | | 2 | .05 | (.00, .51) | | | | | | | | $Q_E$ (2) = 65.58, $p < .01$ |
| | | | | | | | | | | | | | | | | $Q_M$ (2) = 119.76, $p < .01$ |
| Context [c] | | | | | | | | | | | | | | | | |
| Romantic | 5 | .19 | (.09, .35) | | | 2 | .30 | (.27, .34) | | | | | | | | $Q_E$ (5) = 202.64, $p < .01$ |
| | | | | | | | | | | | | | | | | $Q_M$ (2) = 442.69, $p < .01$ |
| Willingly [d] | | | | | | | | | | | | | | | | |
| Voluntary | 5 | .13 | (.07, .23) | | | - | - | - | | | - | - | - | | | Q (4) = 253.01, $p < .01$ |
| Unsolicited | - | - | - | | | 4 | .23 | (.15, .34) | | | - | - | - | | | Q (3) = 31.83, $p < .01$ |
| Without consent | - | - | - | | | - | - | - | | | 2 | .04 | (.02, .06) | | | Q (1) = 7.96, $p < .01$ |
| Sex differences | | | | .61 | .68 | | | | .88 | .77 | | | | .68 | .07 | $Q_E$ (112) = 5355.83, $p < .01$ |
| | | | | | | | | | | | | | | | | $Q_M$ (3) = 3.45, $p = .54$ |
| Women | 31 | .17 | (.13, .21) | | | 21 | .34 | (.26, .41) | | | 8 | .07 | (.05, .10) | | | |
| Men | 30 | .16 | (.13, .20) | | | 20 | .39 | (.31, .47) | | | 8 | .12 | (.09, .16) | | | |
| Mean age | 37 | | | | < .01 | 25 | | | | < .01 | 7 | | | | .05 | $Q_E$ (63) = 193.77, $p < .01$ |
| 12 | | .04 | (.02, .06) | | | | .13 | (.07, .22) | | | | .02 | (.01, .07) | | | $Q_M$ (3) = 148.00, $p < .01$ |

*(Continued)*

**Table 3.** (Continued)

| | Sending | | | | | Receiving | | | | | Forwarding | | | | | |
|---|---|---|---|---|---|---|---|---|---|---|---|---|---|---|---|---|
| | k | eff | (95% CI) | Tau² | p | k | eff | (95% CI) | Tau² | p | k | eff | (95% CI) | Tau² | p | Comparison |
| 14 | | .09 | (.07, .12) | | | | .23 | (.18, .30) | | | | .05 | (.03, .09) | | | |
| 16 | | .21 | (.17, .25) | | | | .39 | (.32, .46) | | | | .10 | (.06, .19) | | | |

"*k*" = number of studies included, "*eff*" = effect size (prevalence), "95% CI" = 95% confidence interval, "*Q*" = Cochran's Q for heterogeneity detection, "$Q_E$" = within-categories statistic to test the model misspecification, "$Q_M$" = between-categories statistic to test the influence of the moderator variable on the prevalence rates, "*Tau²*" = Residual heterogeneity for the levels of the inner factor, "*p*" = p-values for the test statistics.

[a] Prevalence estimates considering the geographical origin of samples were not compared with a significance test, but are provided for descriptive purposes only.

[b] Insufficient "k" to make comparisons.

[c] The context in which sexting was carried out was not specified or was not clearly defined in the rest of the studies.

[d] No studies were found specifying non-voluntariness or the requesting or expression of consent in the experiences of sending, receiving or forwarding sexts.

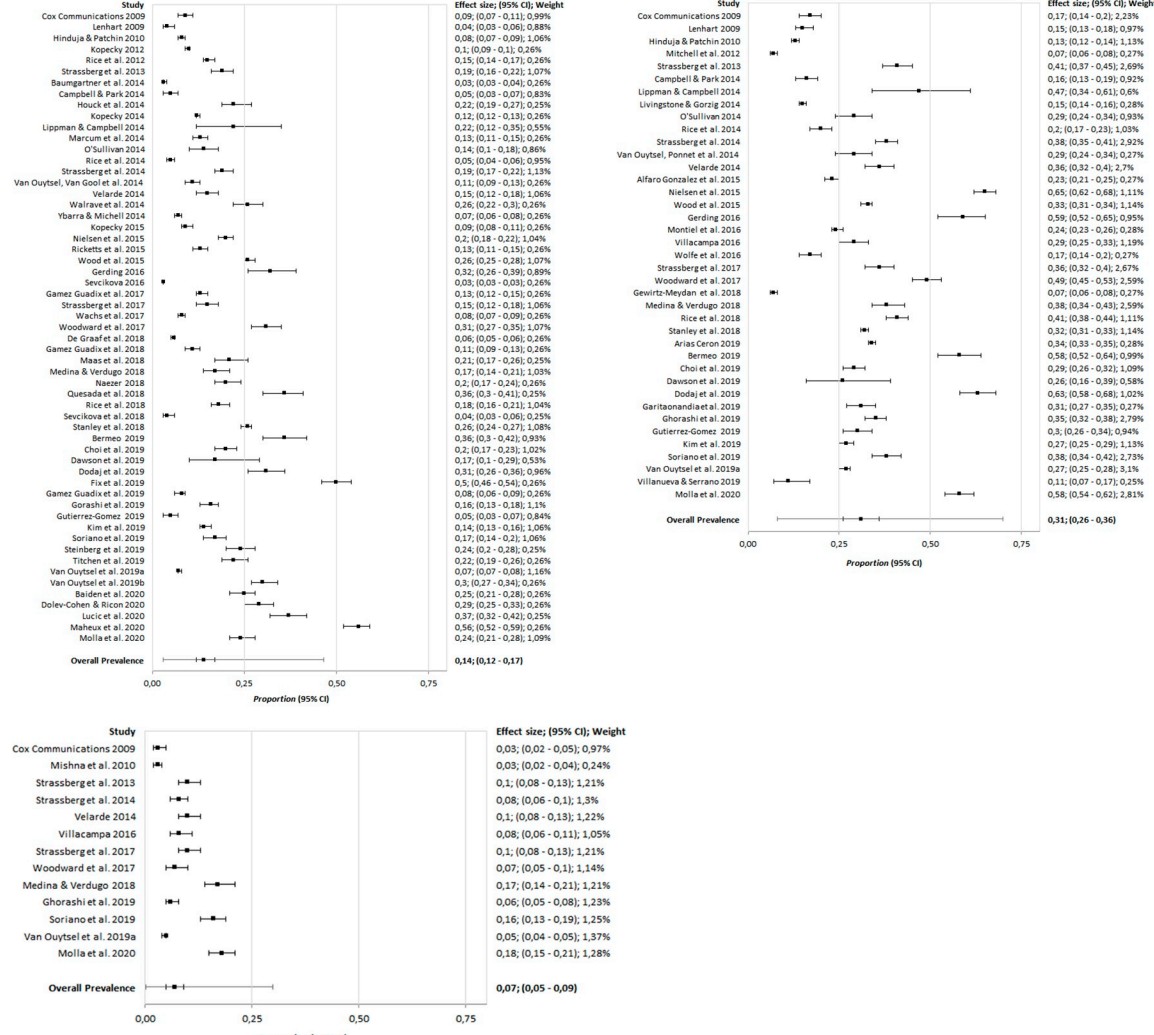

**Fig 2. a.** Forest plot of the observed prevalences and the overall mean estimate of sending sexts. Studies by Van Ouytsel et al. 2019a and Van Ouytsel et al. 2019b correspond to reference numbers 66 and 65 in S1 Appendix. **b.** Forest plot of the observed prevalences and the overall mean estimate of receiving sexts. **c.** Forest plot of the observed prevalences and the overall mean estimate of forwarding sexts.

in prevalence estimates that may, in part, be explained by both methodological and conceptual factors.

The low quality of our meta-analytic sample is an important aspect to highlight in our research, and this aspect has affected the estimated sexting prevalence rates. This research, indeed, identifies that sample representativeness is a significant moderator of prevalence variability, as has already been documented in previous reviews [1]. Our results show that studies with probabilistic samples gave significantly lower prevalences in all sexting experiences. The prevalence estimates of sending, receiving and forwarding sexts in probabilistic samples were significantly lower than the overall prevalences reported in our own study, and also than the overall prevalences reported by Madigan et al. [2]. Although only a small number of studies used random sampling procedures, the value of the selection bias, for example, demographic representation, significantly affects prevalence estimates. The non-representativeness of samples and other characteristics relating to the methodological quality of the studies (as discussed below) may be overestimating the true prevalences of sexting. Regarding sexting measure quality, our assessment also reveals that as many as 72% ($n = 57$) of the reviewed studies did not report any reliability index or evidence of validity. In this respect, results only showed statistical differences among studies classified as low versus significant measurement risk of bias regarding the forwarding of sexts. The non-difference found in the experiences of sending and receiving sexts it is not directly interpretable, since there may be compensatory effects between studies in which bias potentially increased or decreased prevalence rates. Future research should specifically address the reliability or validity of the sexting measures used. Furthermore, results that consider the timeframe of the measure of sexting suggest that responses may be subject to recall bias. Lastly, study sample sizes varied considerably (from 51 to 21372), which may limit the comparability of the studies. All such quality-related aspects reasonably warrant the wide credibility / prediction intervals obtained in our study, and imply that a wide range of values may also be obtained in future observations. On the basis of our results, we recommend that future empirical research study sexting with representative samples, use validated instruments, report on the reliability of obtained responses, and investigate sexting over a short time frame in order to reduce recall bias. Concerning differences in results according to the data collection procedure applied, it was found that the estimated prevalence of receiving sexts varied significantly, in accordance with the hypothesis of Barrense-Dias et al. [15]. In depth analysis of these results shows that the self-reported administration procedure clearly affects the accuracy of estimates, presenting a more homogeneous estimation of prevalence of sending and receiving sexts than face-to-face and telephone interviews. However, it is problematic to compare such results on account of the fact that the employed sampling method also plays a significant role in the accuracy of prevalence estimates.

## Regarding demographic factors, our results lead to the conclusion that no gender

differences appear in any sexting experiences. This finding concurs with Madigan et al. [2] results. The research also suggests that the practice of sexting is more prevalent with increasing age [1, 2]. This result suggests that educational measures in schools to inform pupils of the opportunities (e.g. as a sexual exploration or in order to initiate sexual relationships) and of the risks of sexting (e.g. non-consensual distribution of sexts) should be implemented mainly at early adolescent stages. Regarding conceptual factors, unlike Madigan et al. [2], our results show that the prevalence of sexting is moderated by the type of media content transmitted. Specifically, the sending of text messages obtained significantly higher global prevalence than the sending of pictures or videos. In this regard, sexting may be a gradual evolving activity that

begins with the exchange of text messages and leads to the exchange of other media formats such as images or videos [15]. It is also reasonable to think that the exchange of text messages may require a lower degree of exposure and of trust between the sender and receiver compared to the exchange of images or videos [14]. Segregating the estimates based on media content type, our estimated prevalence for receiving text messages is higher than the overall prevalence estimated by Klettke et al. [1]. Future empirical studies should also consider the content of the messages in terms of the purposes for which they are sent or received (e.g., expressing sexual interest towards the recipient, describing a real or fictional erotic scene, proposing to perform cybersex or to enact live sexual relations). They also should broaden and clearly define the different types of media content exchanged, including text messages, images, videos, and, additionally, audio recordings, which can be considered media content useful to fulfilling a sexual purpose [14, 36], and voice calls of a libidinous character that can be used by individuals to excite or satisfy their own or someone else's sexual pleasure.

Another result to be highlighted is that elements such as the degree of sexual explicitness of the media content, the context in which the sexting is carried out, the willingness of participants and the timeframe of the measure of sexting were not made explicit in the majority of operational definitions reviewed, and were thus left subject to the interpretation of respondents. The lack of definition in elements such as the context and willingness of participants is worrying, because both are key indicators allowing professionals to identify and differentiate between: a) the practice of sexting as a consensual sexual expression activity in the context of a romantic relationship; and b) sexting as a result or consequence of manipulation or coercion. Clarifying these elements in the operational definition of sexting remains a priority for future research on sexting. Researchers should also ascertain whether the faces of participants are visible in the images or videos, since several studies have indicated that the majority of participants indicating sending nude and semi-nude depictions recognized having included their faces [37], and the consequences of the malicious use of pictures or videos in which one is easily identifiable or recognizable may be particularly harmful [14].

Recapitulating, this paper shows how certain conceptual and methodological choices influence prevalence estimates of sexting experiences among youths. Similar operationalizations of sexting [38] and a more detailed report of its defining elements would allow us to more accurately compare the prevalences of sexting and study the causes of its heterogeneity. In a nutshell, consensual methodological procedures must be established for use in both the fieldwork (e.g., sampling techniques, administration procedures) and analysis of sexting (e.g., actions, media content type, explicitness, temporal framework).

## Study limitations

This research is not without its limitations. Difficulties were encountered in extracting information from the studies regarding contextual variables to aid the characterization of sexting, including those relating to sample socio-demographic aspects [39]. Further limitations are the potential selection and measurement bias identified in many of the studies reviewed and the difficulty of synthesizing heterogeneous results on the prevalence of sexting. For example, in 10 studies, prevalence results were incalculable due to the disaggregate form of data, for example, in terms of the channel used for transmitting the sexts (e.g., via cell phone, social network) or relationship type (e.g., peers, online friends, strangers) (see S2 Table). Studies were only selected for inclusion if they provided or facilitated the calculation of a combined estimate of prevalence of sending, receiving or forwarding sexts.

It was also not possible to carry out additional planned comparisons between certain subgroups due to a smaller number of studies assessing such moderators as the degree of sexual

explicitness of the media content, the context in which the sexting is practiced, the willingness of participants or the timeframe of the measure of sexting. For this same reason, it was also not possible to carry out stratified analyses by year of data collection of the moderator effects.

Finally, another limitation of the meta-analysis is that it is inadequately representative of the entire world population. Data from developing countries, from non-occidental countries and from younger (under 12 years of age) were scarce.

## Conclusion

Our meta-analysis results suggest high mean prevalences of sending and receiving sexts involving youths in studies published between 2009 and 2020 (.14, 95% CI: .12, .17, and .31, 95% CI: .26, .36, respectively). Additionally, mean prevalences of sending, receiving and forwarding sexts increased with data collection year (e.g., .07, 95% CI: .05, .10, for sending sexts in studies collecting data in 2009, versus .16, 95% CI: .13, .20 in 2014, and .33, 95% CI: .22, .46 in 2018) and age (e.g., .04, 95% CI: .02, .06, for sending sexts at the age of 12, versus .09, 95% CI: .07, .12, at the age of 14, and .21, 95% CI: .17, .25, at the age of 16, averaging all studies reviewed).

The results also indicate difficulties in accurately determining the prevalence of sexting experiences. In this regard, the high heterogeneity of the meta-analysis results is affected by both methodological and conceptual issues. This paper's results highlight the importance of methodological aspects such as sampling techniques, as probabilistic samples helped to explain the encountered heterogeneity, and led to lower mean prevalence estimates in the global time period studied (.08, 95% CI: .06, .11; .19, 95% CI: .15, .24; and .04, 95% CI: .03, .07; for sending, receiving and forwarding sexts, respectively). Self-reported administration procedures (e.g., paper and online questionnaires) also led to more homogeneous prevalence estimates than interview methods (e.g., face-to-face or telephone interviews). Furthermore, the prevalence of forwarding sexts varied slightly according to the timeframe of the measure. Regarding conceptual factors, media content type also moderated the prevalence of sexting, with text messages transmitted more frequently (e.g., .22, 95% CI: .18, .27, for sending sexts) than images or videos (.12, 95% CI: .10, .15), averaged across all the studies analyzed. In this sense, future efforts should carefully explore the content of the text messages exchanged, which is how the practice of sexting appears to begin. Finally, high heterogeneity in prevalence estimates together with the significant risk of bias observed in many of the synthesized studies underscore the need for greater consensus on the definition of sexting. Nevertheless, we believe that the results obtained do make a valuable contribution to the advancement of research on sexting, and provide arguments to guide new studies on the subject, proposals for more suitable definitions of sexting, and more reliable and valid measurement procedures.

## Supporting information

**S1 Checklist. PRISMA 2009 checklist.**
(DOCX)

**S1 Table. Search strategy used.** Subject areas excluded: Business, Management and Accounting, Economics, Econometrics and Finance, Biochemistry, Genetics and Molecular Biology and Pharmacology, Toxicology and Pharmaceutics.
(DOCX)

**S2 Table. Excluded studies.**
(DOCX)

**S3 Table. Critical appraisal of the studies.** "HDHD" = attention-deficit/hyperactivity disorder, "na" = Not available, "Yes" = The study provides minimum and maximum age, "Not

completely" = Provides at least minimum, maximum or average age, "No" = Does not provide any data.
(DOCX)

**S4 Table. Characteristics of included studies.** "DT-NPR" = degree thesis no peer-reviewed, "MT-NPR" = master thesis no peer-reviewed, "PT-NPR" = published thesis no peer-reviewed, "R-NPR" = report no peer-reviewed, "A-PR" = article peer-reviewed, "Cs" = cross sectional, "L" = longitudinal, "P" = probabilistic, "NP" = no probabilistic, "Os" = online survey, "Ps" = paper survey, "O&Ps" = online and paper survey, "TI" = telephone interview, "PI" = personal interview, "Mo" = Devices monitorization, "DC" = disaggregated by contents, "na" = not available, insufficient information or unclear.
(DOCX)

**S1 Appendix. References included in the meta-analysis.**
(DOCX)

**S2 Appendix. Relevant R code and graphs.**
(DOCX)

## Author Contributions

**Conceptualization:** Cristian Molla-Esparza, Emelina López-González.

**Data curation:** Cristian Molla-Esparza, Josep-Maria Losilla, Emelina López-González.

**Formal analysis:** Cristian Molla-Esparza, Josep-Maria Losilla.

**Funding acquisition:** Josep-Maria Losilla.

**Methodology:** Josep-Maria Losilla, Emelina López-González.

**Project administration:** Josep-Maria Losilla.

**Software:** Cristian Molla-Esparza, Josep-Maria Losilla.

**Supervision:** Josep-Maria Losilla, Emelina López-González.

**Validation:** Josep-Maria Losilla, Emelina López-González.

**Visualization:** Cristian Molla-Esparza.

**Writing – original draft:** Cristian Molla-Esparza.

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
