## [Decision Letter · Decision Letter 0]

2 Nov 2020

PONE-D-20-25644

Sexting Prevalence and its Heterogeneity in Samples of Minors: A Three-level Meta-analysis with Multiple Outcomes

PLOS ONE

Dear Dr. Molla Esparza,

Thank you for submitting your manuscript to PLOS ONE. After careful consideration, we feel that it has merit but does not fully meet PLOS ONE’s publication criteria as it currently stands. Therefore, we invite you to submit a revised version of the manuscript that addresses the points raised during the review process.

We look forward to receiving your revised manuscript.

Kind regards,

Angelo Brandelli Costa

Academic Editor

PLOS ONE

Journal Requirements:

Reviewers' comments:

Reviewer's Responses to Questions

**Comments to the Author**

1. Is the manuscript technically sound, and do the data support the conclusions?

Reviewer #1: Yes

Reviewer #2: Yes

2. Has the statistical analysis been performed appropriately and rigorously? 

Reviewer #1: Yes

Reviewer #2: Yes

3. Have the authors made all data underlying the findings in their manuscript fully available?

Reviewer #1: Yes

Reviewer #2: Yes

4. Is the manuscript presented in an intelligible fashion and written in standard English?

Reviewer #1: Yes

Reviewer #2: Yes

5. Review Comments to the Author

Reviewer #1: I was satisfied of reading the manuscript “Sexting Prevalence and its Heterogeneity in Samples of Minors: A Three level Meta-analysis with Multiple Outcomes”. I do have quite a few recommendations. There are subtle, but important aspects that need to be corrected.

- I think the title could be improved. For instance, “its heterogeneity” adds little to the title. “Samples of Minors” is quite a redundancy. Perhaps be specific about the age group investigated? In the introduction, there is a mixture of terms: minors, adolescents, juvenile. Be specific as possible.

- Using “multiple outcomes” in the title is kind of mysterious. If I am searching for your paper, how do I know if the full text contains health, school, psychological outcomes? I think that deserves a careful thought.

- I am not happy with the abstract. I guess it does not present the goal and methodology accordingly. I would avoid presenting statistics in the abstract (e.g., .22, 95% CI: .18, .27 for sending text messages, versus .12, 95% CI: .10, .15 for images or videos). Could you make sense of these statistics for your reader?

- I am not sure that the last sentence of the abstract really concludes what was presented (“Overall, the results underline the need to seek further consensus on the definition and measurement of sexting”). Again, I think this bit deserves a careful thought.

- In the introduction, it is confusing the way prevalence is presented. What is .10? Is it 10%? What did Klettke and collaborators find? In the same sentence, what is “with a high heterogeneity in the results” and why is that important? These same comments apply to all further descriptions of previous meta-analyses in the introduction.

- Still in the introduction, line 60 “Despite progress achieved in the study of sexting prevalence”. What kind or progress? This is novel information, and I am curious to know what has been achieved. Surprisingly, authors claim to advance the field but conclude basically what others have done already? (“previous review studies on the topic have concluded that the main difficulty in accurately determining the prevalence of sexting lies in the lack of consensus on such a fundamental issue as the actual operational definition of sexting”, lines 61-63, and similarly when discussing the results [lines 444 and so on]).

- Not sure you need to mention the year of the last meta-analysis: “therefore, the first aim of this research was to update the previous meta-analytic synthesis on sexting prevalence among juveniles, including studies up until 2016”.

- What is “applied administration” (line 77)? Also: sample study quality? Do you mean sampling robustness?

- Do you think that not having independent researchers searching and extracting data biased your study? Some sort of justification/explanation is necessary.

- Why it has not been registered prior to data collection? Some sort of justification/explanation is necessary.

- The choice for geographical region is unusual. I would be surprised to see a publication from Antarctica, for instance. However, regional differences in Europe and even in South America are expected due to cultural aspects.

- The supplementary material S1 incorrectly says that registration was identified in the abstract. Please, review the guidelines mentioned in S1.

- Please, do not use quoted sentences/direct citations in scientific writing (lines 408-409).

- The insertion of the Ecological Momentary Assessment comes out of the blue, with little justification why it would reduce bias. Either explain it or exclude it.

- I am not sure this is novel information and deserved to be in the paper “Our findings indicate that best evidence synthesis in the study of sexting prevalence should be guided by studies that mainly apply random sampling, with a non-significant proportion of non-responses and use of validated instruments in the study sample, or in comparable samples”.

- This is a bit grandiose. I would soften it “This systematic review and meta-analysis research exhaustively explores available results on the prevalence of sexting, examining a greater number and more diverse set of empirical studies on sexting prevalence than previous reviews [1,2,4], and estimating the prevalence of sexting experiences via three-level, mixed-effects, meta-analysis models” (line 320).

Reviewer #2: In Title, line 6.

The term "Minor" used in the title of the article refers to minors legally. The sample of the article covers people youth under 19, however, the legal age of majority is different according to the legislation of each country. I suggest that the term be changed to “Juventos”, “Youngsters”, "Youth people" or the term that the authors deem more appropriate for the development phase that the article covers.

In Introduction, line 73.

The authors argue in the introduction that sexting may be a growing risky behavior, but they do not use any reference or justification for it. What risks? Is there evidence that this behavior causes emotional and / or social damage? To elaborate, perhaps a short paragraph, about the current public and scientific debate about sending sexting and the impact on the social development of these young people. It is important to contextualize the reader of the social impact of this behavior to justify the need for a study on the theme.

In Methodology, line 13, first paragraph.

This paragraph with the definition is already described in the introduction (line 44). It does not need to be repeated.

In Discussion.

Line 400

“This result suggests that educational measures in schools to inform pupils of the risks of sexting should be implemented mainly at early adolescent stages”.

Again, what risks are the authors referring to that justify an early educational intervention? Elaborate in the introduction of the article.

Line 426.

“Considering potential negative consequences of sexting [34]”

This reference to another study is very vague. The authors of article [34] mentioned in the sentence refer mainly to Sextortion. It becomes clearer and more interesting for the reader if at least this information is in the sentence. If necessary, including another risk.

Example: Sextortion has been identified as an emerging online threat to youth who send sexting [34]. Considering this potential negative consequence of sexting (…).

(It is only an example, to reformulate in the way that the authors find more coherent).

6. PLOS authors have the option to publish the peer review history of their article (what does this mean?). If published, this will include your full peer review and any attached files.

Reviewer #1: **Yes: **Dr. Guilherme Welter Wendt

Reviewer #2: No

---

## [Author Response · Author response to Decision Letter 0]

18 Nov 2020

Response to Reviewers

Reviewer #1. I was satisfied of reading the manuscript “Sexting Prevalence and its Heterogeneity in Samples of Minors: A Three level Meta-analysis with Multiple Outcomes”. I do have quite a few recommendations. There are subtle, but important aspects that need to be corrected.

Authors’ response: The authors appreciate the time and effort you have dedicated to providing valuable suggestions for our manuscript. All of them have been answered in a reasoned manner and the manuscript has been revised accordingly. Consequently, we believe that the contents and the clarity of our paper are much improved in the revised version.

Comment 1 and 2 on the Title. [Comment 1] I think the title could be improved. For instance, “its heterogeneity” adds little to the title. “Samples of Minors” is quite a redundancy. Perhaps be specific about the age group investigated? In the introduction, there is a mixture of terms: minors, adolescents, juvenile. Be specific as possible. [Comment 2] Using “multiple outcomes” in the title is kind of mysterious. If I am searching for your paper, how do I know if the full text contains health, school, psychological outcomes? I think that deserves a careful thought.

Authors’ response. We are very grateful for your reflections on this matter. We have decided to simplify the title of the manuscript as “Prevalence of Sending, Receiving and Forwarding Sexts among Youths: A Three-Level Meta-Analysis” (Line 5). We have changed the term "minors" to the term “youths” in the title and throughout the manuscript. We have also eliminated the term “heterogeneity”, because considering heterogeneity is an inherent part of any meta-analytical study. Lastly, we have also replaced “multiple outcomes” with the specific sexting experiences we have analyzed: sending, receiving and forwarding of sexts.

Comment 3 and 4 on the Abstract. [Comment 3] I am not happy with the abstract. I guess it does not present the goal and methodology accordingly. I would avoid presenting statistics in the abstract (e.g., .22, 95% CI: .18, .27 for sending text messages, versus .12, 95% CI: .10, .15 for images or videos). Could you make sense of these statistics for your reader? [Comment 4]. I am not sure that the last sentence of the abstract really concludes what was presented (“Overall, the results underline the need to seek further consensus on the definition and measurement of sexting”). Again, I think this bit deserves a careful thought.

Authors’ response: We appreciate that you have asked for more accurate wording in the abstract. Following your suggestion, we have clarified the objective and the methodology of the study, and have presented the results in a clearer way. We have also changed the last sentence of the abstract to conclude with a statement about the high and increasing prevalence of sending and receiving of sexts among youths (Lines 24 to 43).

Comment 5 on the Introduction. In the introduction, it is confusing the way prevalence is presented. What is .10? Is it 10%? What did Klettke and collaborators find? In the same sentence, what is “with a high heterogeneity in the results” and why is that important? These same comments apply to all further descriptions of previous meta-analyses in the introduction. 

Authors’ response: We agree with your comment. As requested, we have expanded the information on the findings of the studies of Klettke et al. and Madigan et al., and have modified and explained the aspects you have indicated to us (Lines 61 to 80. In addition, in the Method section, we have clarified that meta-analysis results are out of 1 (Line 205).

Comment 6 on the Introduction. Still in the introduction, line 60 “Despite progress achieved in the study of sexting prevalence”. What kind or progress? This is novel information, and I am curious to know what has been achieved. Surprisingly, authors claim to advance the field but conclude basically what others have done already? (“previous review studies on the topic have concluded that the main difficulty in accurately determining the prevalence of sexting lies in the lack of consensus on such a fundamental issue as the actual operational definition of sexting”, lines 61-63, and similarly when discussing the results [lines 444 and so on]).

Authors’ response: Thank you for pointing this out. This expression has been removed from the introduction. Its wording erroneously stated that previous reviews had reached this conclusion, when, in fact, they only raised it as a working hypothesis.

This working hypothesis is the one we have empirically tested in our meta-analysis. Consequently, the conclusion that the difficulty in determining the prevalence of sexting is partly explained by the variety of definitions used in primary studies is derived from the results of our meta-analysis. This conclusion, together with our proposal to work towards a greater consensus on the definition of sexting, is now presented only at the end of the discussion (L 447 to L 451).

Comment 7 on the Introduction. Not sure you need to mention the year of the last meta-analysis: “therefore, the first aim of this research was to update the previous meta-analytic synthesis on sexting prevalence among juveniles, including studies up until 2016”.

Authors’ response: In accordance with your suggestion, we have eliminated the year of the last meta-analysis.

Comment 8 on the Introduction. What is “applied administration” (line 77)? Also: sample study quality? Do you mean sampling robustness?

Authors’ response: In accordance with your comment, and, since both terms are not precise, we have replaced “sample study quality” by “sampling techniques” and “applied administration” by “administration procedure used” (Line 97). We have also checked that this terminology is consistent throughout the manuscript.

Comment 9 and 10 on the Methodology. [Comment 9] Do you think that not having independent researchers searching and extracting data biased your study? [Comment 10] Why it has not been registered prior to data collection? Some sort of justification/explanation is necessary.

Authors’ response: Thank you for pointing this out. To facilitate the replication of our review, in the last paragraph of the section “Search and selection of documents” we have referenced Table S1 (Lines 128 to 130), which contains the specific search strategy used in each database consulted. We have also added a second paragraph in the section “Inclusion and exclusion criteria”, with a detailed description of the selection procedure applied (Lines 138 to 142).

Comment 11 on the Methodology. The choice for geographical region is unusual. I would be surprised to see a publication from Antarctica, for instance. However, regional differences in Europe and even in South America are expected due to cultural aspects.

Authors’ response: We agree fully with your reflection. We have added the report of the prevalence of sexting by country when we have enough studies to estimate it (Table 3). Furthermore, in the supplementary material we have included Table S4 B, with the classification, by continent and country, of the number of samples of studies included in our review. 

Comment 12 on the Supplementary Files. The supplementary material S1 incorrectly says that registration was identified in the abstract. Please, review the guidelines mentioned in S1.

Authors’ response: We have modified the Abstract cell of the PRISMA Guidelines by including the following sentence “Page 2 (no protocol was registered)”. Likewise, due to the modifications made throughout the manuscript, we have also revised the other sections/topics.

Comment 13 on the Discussion. Please, do not use quoted sentences/direct citations in scientific writing (lines 408-409).

Authors’ response: In accordance with your comment, we have rewritten this sentence as “Concerning differences in results according to the data collection procedure applied, it was found that the estimated prevalence of receiving sexts varied significantly, in accordance with the hypothesis of Barrense-Dias et al. [15].” (Line 397 to 400). 

Comment 14 on the Discussion. The insertion of the Ecological Momentary Assessment comes out of the blue, with little justification why it would reduce bias. Either explain it or exclude it.

Authors’ response: Thank you for your comment on this matter. We agree with you that this proposal for the evaluation of sexting is not derived from the results of our review. We have, therefore, decided to exclude it.

Comment 15 on the Discussion. I am not sure this is novel information and deserved to be in the paper “Our findings indicate that best evidence synthesis in the study of sexting prevalence should be guided by studies that mainly apply random sampling, with a non-significant proportion of non-responses and use of validated instruments in the study sample, or in comparable samples”.

Authors’ response: We have modified the wording according to the results obtained. Our revision thus indicates that the prevalence of sexting is moderated by the sampling technique (probabilistic vs non-probabilistic), the administration procedure (interview vs self-reported), the quality of the measure, and the time frame asked. See Lines 394 to 397.

Comment 16 on the Discussion. This is a bit grandiose. I would soften it “This systematic review and meta-analysis research exhaustively explores available results on the prevalence of sexting, examining a greater number and more diverse set of empirical studies on sexting prevalence than previous reviews [1,2,4], and estimating the prevalence of sexting experiences via three-level, mixed-effects, meta-analysis models” (line 320).

Authors response. In accordance with your comment, we have rewritten this sentence as “This systematic review and meta-analysis research examines the prevalence of sexting experiences via three-level, mixed-effects, meta-analysis models” (Lines 344 to 348).

Reviewer #2

Authors’ comment: We are grateful for the valuable suggestions provided. All of them have been considered and the manuscript has been revised accordingly.

Comment 1 on the Title. In Title, line 6. The term "Minor" used in the title of the article refers to minors legally. The sample of the article covers people youth under 19, however, the legal age of majority is different according to the legislation of each country. I suggest that the term be changed to “Juventos”, “Youngsters”, "Youth people" or the term that the authors deem more appropriate for the development phase that the article covers.

Authors’ response: Thank you for pointing this out. Indeed, the term “minor” describes someone who is still legally considered a child, and the legal age of majority varies depending on the country or state of residence. We believe that “youth” is the term that better describes our reference population. We have modified the wording of the title and we have changed the term “minors” to “youths” through the manuscript. 

Comment 2, 4, and 5 on sexting opportunities and risks.

[Comment 2] In Introduction, line 73. The authors argue in the introduction that sexting may be a growing risky behavior, but they do not use any reference or justification for it. What risks? Is there evidence that this behavior causes emotional and / or social damage? To elaborate, perhaps a short paragraph, about the current public and scientific debate about sending sexting and the impact on the social development of these young people. It is important to contextualize the reader of the social impact of this behavior to justify the need for a study on the theme. 

Authors’ response: We appreciate your suggestion. To address these issues, we have written a short paragraph on the implications and possible consequences of sexting, as well as the psychosocial problems that the empirical literature has identified. The paragraph can be found in lines 47 to 58.

[Comment 4]. In Discussion. Line 400. “This result suggests that educational measures in schools to inform pupils of the risks of sexting should be implemented mainly at early adolescent stages”. Again, what risks are the authors referring to that justify an early educational intervention? Elaborate in the introduction of the article. 

Authors’ response: Having added a short paragraph about opportunities and risks of sexting in the Introduction, we have reformulated the sentences mentioning such risks and given examples (Lines 409 to 412).

[Comment 5] Line 426. “Considering potential negative consequences of sexting [34]” This reference to another study is very vague. The authors of article [34] mentioned in the sentence refer mainly to Sextortion. It becomes clearer and more interesting for the reader if at least this information is in the sentence. If necessary, including another risk. Example: Sextortion has been identified as an emerging online threat to youth who send sexting [34]. Considering this potential negative consequence of sexting (…). (It is only an example, to reformulate in the way that the authors find more coherent).

Authors’ response: Thank you for the comment. We have decided to remove this phrase since it does not specifically support the argument that precedes it.

Comment 3 on the Methodology. In Methodology, line 13, first paragraph. This paragraph with the definition is already described in the introduction (line 44). It does not need to be repeated.

Authors’ response: Thank you for pointing this out. Indeed, it is redundant. Accordingly, we have removed the definition of sexting that we had included at the beginning of the Methodology section.

---

## [Editor Report · Decision Letter 1]

25 Nov 2020

Prevalence of Sending, Receiving and Forwarding Sexts among Youths: A Three-Level Meta-Analysis

PONE-D-20-25644R1

Dear Dr. Molla Esparza,

We’re pleased to inform you that your manuscript has been judged scientifically suitable for publication and will be formally accepted for publication once it meets all outstanding technical requirements.

Kind regards,

Angelo Brandelli Costa

Academic Editor

PLOS ONE
---

## [Editor Report · Acceptance letter]

27 Nov 2020

PONE-D-20-25644R1 

Prevalence of Sending, Receiving and Forwarding Sexts among Youths: A Three-Level Meta-Analysis 

Dear Dr. Molla-Esparza:

I'm pleased to inform you that your manuscript has been deemed suitable for publication in PLOS ONE. Congratulations! Your manuscript is now with our production department. 

Kind regards, 

on behalf of

Dr. Angelo Brandelli Costa 

Academic Editor

PLOS ONE